# Spatial–Temporal Variability of the Calculated Characteristics of the Ocean in the Arctic Zone of Russia by Using the NEMO Model with Altimetry Data Assimilation

**Konstantin Belyaev [1,2], Andrey Kuleshov [3,*] and Ilya Smirnov [4]**

[1]  Shirshov Institute of Oceanology, Russian Academy of Sciences, 117997 Moscow, Russia; kosbel55@gmail.com

[2]  Federal Research Center "Computer Science and Control", Russian Academy of Sciences, 119333 Moscow, Russia

[3]  Keldysh Institute of Applied Mathematics, Russian Academy of Sciences, 125047 Moscow, Russia

[4]  Lomonosov Moscow State University, Faculty of Computational Mathematics and Cybernetics, 119991 Moscow, Russia; ismirnov@cs.msu.ru

*  Correspondence: andrew_kuleshov@mail.ru; Tel.: +7-499-220-0722

**Abstract:** The spatial–temporal variability of the calculated characteristics of the ocean in the Arctic zone of Russia is studied. In this study, the known hydrodynamic model of the ocean Nucleus for European Modelling of the Ocean (NEMO) is used with assimilation of observation data on the sea surface height taken from the Archiving, Validating and Interpolation Satellite Observation (AVISO) archive. We use the Generalized Kalman filter (GKF) method, developed earlier by the authors of this study, in conjunction with the method of decomposition of symmetric matrices into empirical orthogonal functions (EOF, Karhunen–Loeve decomposition). The investigations are focused mostly on the northern seas of Russia. The main characteristics of the ocean, such as the current velocity, sea surface height, and sea surface temperature are calculated with data assimilation (DA) and without DA (the control calculation). The calculation results are analyzed and their spatial–temporal variability over a time period of 14 days is studied. It is shown that the main spatial variability of characteristics after DA is in good agreement with the localization of currents in the North Atlantic and in the Arctic zone of Russia. The contribution of each of the eigenvectors and eigenvalues of the covariation matrix to the spatial–temporal variability of the calculated characteristics is shown by using the EOF analysis.

**Keywords:** model of ocean circulation NEMO; GKF data assimilation method; Arctic zone of Russia

## 1. Introduction

In this study, we continue a cycle of investigations on development and application of data assimilation (DA) methods in the models of ocean dynamics [1–3]. There is a relatively small number of works devoted especially to DA in the Arctic. This can be explained by both the insufficiency of the database of observations in the Arctic zone and the complexity of the ocean dynamics because of the ice cover, complexity of the bottom topography, tidal forces, and other factors. There are several numerical models that try to capture the Arctic dynamics and its structure; however, none of them describe completely the physical processes that really occur in this region. Therefore, it is reasonable to use a more sophisticated and developed DA technique to discover special features in the Arctic zone. The ocean dynamics in the Arctic zone is simulated with the help of the known hydrodynamic model Nucleus for European Modelling of the Ocean (NEMO) developed in the Institute Pierre Simon Laplace

in Paris and described in several papers (see, for example, [4,5]) with application of the DA method developed earlier by the authors of this study, namely, the Generalized Kalman filter (GKF) [1–3]. The GKF method is the generalization of the known data assimilation method—the Ensemble Optimal Interpolation (EnOI) method, and has some advantages over this method, which is shown, in particular, in [2]. The GKF method was earlier used together with the known model of ocean dynamics HYbrid Circulation Ocean Model (HYCOM) [6] and the model created in the Marchuk Institute of Numerical Mathematics of the Russian Academy of Sciences [7,8]. In our current study, this method is applied in conjunction with the NEMO model of ocean circulation. The main attention is focused on studying the hydrodynamics of the northern seas of Russia as a region of considerable interest from the point of view of physical oceanography, and of economic importance, for which numerous studies are being carried out at the present time (see, for example, [9]).

One of the most important directions in simulation of the ocean and atmosphere dynamics is development and application of DA methods. The main goal of DA is to correct the numerical solution of the dynamics equations by using independent observational data with the purpose of increasing the accuracy of the simulation and forecast of the physical processes under study. Taking into account the progress in both the numerical simulation of such systems with the help of modern high-performance computers and the development of observation instruments, environment monitoring systems, databases, and methods for certainty estimation and rejection of observations, the development of DA methods becomes one of the most relevant directions in physical oceanography, physics of atmosphere, climatology, ecology, and some other geosciences. In meteorology and physical oceanography, the observation data are assimilated into numerical models with the purpose of correcting the results of simulations in the process of calculations to enhance the quality of calculations of the atmosphere and ocean dynamics, weather forecast, predictions of climate changes, as well as to improve the models themselves and the ways of setting the initial and boundary conditions [10,11].

DA is a complex problem that is solved by using methods from different domains of science. When developing the mathematical methods for DA, we use different approaches: variational, stochastic, optimal control, and some other methods. For instance, nowadays, a widely used DA scheme is the 4D-VAR variational method (see, for example, [12,13]). In this method, which was developed by G.I. Marchuk and his disciples [7], the DA problem is reduced to solving the inverse problem of finding the initial condition, starting from which the trajectory of the model solution will pass at "the closest distance" from the observed values in the sense of the given metric. Another widely used approach for solving the DA problem is the stochastic method, in which the sought unknown trajectory of the model system of equations is represented as the sum of a numerical solution, and a random noise and the problem consists in finding the optimal filter. One of the main DA methods is the Ensemble Kalman Filter (EnKF) [14] and its simplified version EnOI, which are often used in different theoretical and applied studies [14–16]. There are the hybrid methods, for example, [17], in which the optimal filtration problem is solved based on minimizing some functional, i.e., as a variational problem. The method we have developed [1–3] belongs to such hybrid methods.

In addition to the purely mathematical approaches to solving DA problems, there are the methods of information technologies, parallel programming, archiving, and fast access to databases, as well as visualization of results and high-speed data transmission. The geographic localization of observation data, their density, observation frequency, their reliability, and noisiness are also of principal importance. All of these aspects make the DA problem extremely difficult and complex; its solution depends on engagement and coordination of efforts of specialists working in various scientific domains.

All of the above-described DA methods are applied in modern calculation models of ocean dynamics. Nowadays, there are a lot of such models developed all over the world whose systems of equations include, as a rule, the Navier–Stokes equations in different approximations, as well as the heat and salt transfer equations. In particular, in many models, the Boussinesq approximation is used, i.e., the medium is assumed to be incompressible; the hydrostatic approximation is also often used [18]. At that, when solving dynamics equations, the calculated characteristics of the model

change continuously in time and space, in the sense that, at small time and space steps, the changes in the calculated characteristics are small.

When using DA, all calculated characteristics of the model change simultaneously, step-wise. In this case, the property of continuity is, as a rule, not satisfied; fictive flows and waves occur. For example, it is shown in [19] which waves may appear when replacing the dynamic fields of calculated characteristics of the model by those obtained as a result of DA. In the same paper, the method is proposed for smoothing (damping) the fictitious waves obtained. Nevertheless, the presence of such fictitious disturbances and their dynamics may lead to a fortiori incorrect results and destroy the calculation using the model. In this connection, it is required to use such a DA method, which would minimize the value of such jumps and, ideally, exclude them at all. At that, it is important not to violate the conservation laws implied in the model equations, which must be satisfied when developing the numerical algorithms and program realization of the model.

Different authors used several DA methods with the NEMO model, in particular, the 4D-VAR method [9]; however, the problem concerning continuity of model characteristics after DA was not investigated. Nowadays, an important direction in physical oceanography is the project whose aim is to provide the analysis and prognosis of the state of the ocean in its particular regions. For this purpose, different mathematical models of the ocean dynamics are used with observation data assimilation and application of different DA methods. For example, in the Brazilian project REMO [20,21], for simulation of the ocean in the regions of production and transportation of hydrocarbons at the shelf of Brazil, the HYCOM model is used for the benefit of the Brazilian state-owned company Petrobras [6]. There are similar projects in Norway (the project TOPAZ [22]), Australia (BlueLink [23]), USA (HYCOM + NCODA [24]), and in other countries. For Russia, the Arctic seas represent a particular interest. Nowadays, numerous and various investigations are carried out there. This region is much less investigated than, for example, the North Atlantic or the Mediterranean Basin, though this region is of great importance for investigation of the universal climatic variability and anthropogenic impact on the environment, and has great economic significance. It should be noted that the development and application of new methods for studying the ocean are of particular importance, where observation data are insufficient, and where absence or lack can be compensated by development of more perfect methods.

In this study, the behavior of the ocean characteristics is investigated by using the NEMO model in conjunction with the authors' DA method in the northern seas of Russia. The main goal of this study is to estimate the influence of altimetry DA and, based on this, to correct the calculation characteristics for more accurate simulation of the ocean dynamics for this region. In addition, in this work, the analysis of the spatial–temporal structure of the fields obtained, in particular, the sea surface height fields, is carried out. Based on the expansion of the transfer functions (matrices) from observations to the model into the EOFs (Karhunen-Loeve decomposition [25,26]), we analyze both the spatial contribution of separate observations and their behavior in time. The EOF expansion is a well-known method in geophysics; however, it is not often used for DA. Nevertheless, using this method, one can determine both the local peculiarities of the DA process (the phase, see Section 3 below) and the energetics of this process (the amplitude) by analogy with the Fourier transform in wave physics.

In this study we have made the following:

- The current sea surface height (SSH), and sea surface temperature (SST) fields are constructed according to the NEMO model before and after DA; their comparative analysis is carried out.
- The variability of these fields for 14 days is shown with consideration of their regional peculiarities.
- Comparison of the constructed model fields before and after DA with the observed fields is performed; their comparative analysis is carried out.
- Based on the EOF expansion, the analysis of the spatial–temporal structure of the SSH fields is carried out; the contribution to the resulting model fields after DA is shown.

We have installed and adapted the NEMO software package to the K-60 Russian high-performance computer [27] in the Keldysh Institute of Applied Mathematics of the Russian Academy of Sciences; the DA program unit is realized according to the GKF method [1].

## 2. Configuration of the General Circulation Model and Data Assimilation Method

The NEMO mathematical model of the general ocean circulation is based on numerically solving the hydrodynamics equations with the continuity equation and the state equation, defined by the UNESCO formula [28]. The system of equations of the NEMO model and the software package are described in detail in [4,5]; therefore, they are not presented in this paper.

For concrete calculations, it is important to determine the model configuration, set the initial and boundary conditions, and the external forcing determined by the atmosphere action, i.e., the action of the wind and heat flows at the ocean-atmosphere interface. The NEMO model is destined to model the entire World Ocean; however, the calculations are performed in the region with coordinates from 60° north to 75° north latitudes and from 0° east to 90° east longitudes. This region includes the areas of the Arctic zone of Russia and the Northeast Passage, in which the influence of the North Atlantic Current is pronounced, but excludes the poorly reproducible ice-field areas located northward of the 75° parallel.

The pitch of the calculation grid is 0.25° in the latitudinal and longitudinal directions. The vertical discretization of the model is 50 levels from the sea surface to the sea bottom with a higher resolution at the levels located closer to the surface (down to a depth of 500 m). The boundary conditions are global; i.e., for the entire region of modeling, at the ocean-land interfaces, we set the conditions of fixed values of temperature and salinity and the zero values of normal derivatives of current velocities (the non-percolation conditions).

The model equations are integrated from the zero initial conditions for the current velocity; the initial and boundary values at the coastline and at the sea bottom for temperature and salinity are taken from the Climate Atlas [29]. At the moving upper ocean-atmosphere boundary, the heat flows and the friction force conditioned by the wind action are set for each climatic month from the NCEP/NCAR atlas [30]. At the initial stage, numerical simulation is performed on a time interval of 40 years (the so-called model spin-up procedure), then, the fields of the restart parameters obtained are used for the further calculation with DA. The spin-up procedure requires considerable computational resources; therefore, in this study we used the restart data [4]. In setting the atmosphere forcing, the DRAKKAR data [5] were used.

The GKF data assimilation method developed by the authors of this study is described in detail in [1,2], and is realized according to the following algorithm:

$$X_{b,n+1} = X_{a,n} + \Lambda(X_{a,n})\Delta t, \tag{1}$$

$$X_{a,n+1} = X_{b,n+1} + K_{n+1}(Y_{n+1} - HX_{b,n+1}), \tag{2}$$

$$K_{n+1} = (\sigma^2_{n+1})^{-1}(\Lambda_{n+1} - C_{n+1})(H\Lambda_{n+1})^{\mathrm{T}}Q^{-1}_{n+1}, \tag{3}$$

$$\sigma^2_{n+1} = (H\Lambda_{n+1})^{\mathrm{T}}Q^{-1}_{n+1}(H\Lambda_{n+1}), \tag{4}$$

where $X_{a,n}$, $X_{b,n}$, $n = 0, 1, \ldots, N$ are the model characteristics' fields at the time step with a number $n$, $n = 0, 1, 2, \ldots$ before and after correction, respectively (the background and analysis states, respectively); $\Lambda$ is the model operator; $\Lambda(X_{a,n}) = \Lambda_{n+1}$ are the characteristics of the model calculation at the step $n+1$; $\Delta t$ is the time step; $Y_n$ is the vector of observed parameters at the step $n$; $K_n$ is the weighting matrix; $H$ is the projection operator (matrix) that projects the space of model states onto the space of observations; i.e., when projecting the values of parameters of the model calculation, these values are

linearly extrapolated to the observation points, and, at that, all of the unobserved model parameters are excluded. The vector $C_n$ is calculated by using the formula

$$C_{n+1} == \frac{M^{-1} \sum_{i=1}^{M} (\hat{X}_{n+1}^i - \hat{X}_n^i)}{\Delta t}. \tag{5}$$

over the grid domain of the model. For that, the Monte Carlo method is used to create the ensemble $\hat{X}_n^i$, $i = 1, 2, \ldots, M$ of $M$ independent model calculations with different initial conditions. The matrix $Q$ is calculated by using the formula

$$Q_{n+1} = M^{-1} \sum_{i=1}^{M} (H\hat{X}_{n+1}^i - HX_n)(H\hat{X}_{n+1}^i - HX_n)^{\mathrm{T}}. \tag{6}$$

As is seen from formulas (1)–(4), the sequence of the analysis fields $X_{a,n}$ is constructed for all $n = 1, 2, \ldots$ ; for this, it is sufficient to have the initial condition $X_{a,0}$, the constructed ensemble fields $\hat{X}_n^i$, $i = 1, 2, \ldots, M$, and observation fields $Y_n$ for all $n$.

It should be noted that this method has one more advantage. It was shown in [1] that at sufficiently small $\Delta t$ and small values of error proportional to $\Delta t$, $Y_{n+1} - HX_{b,n+1}$ will be continuous; the calculated characteristics $X_{a,n+1}$ will preserve the property of continuity in the sense of small changes in the values in time and in space. This means that DA will not lead to significant changes in the characteristics of the process.

Earlier, we have developed a computer programming code for DA by the GCF method used in parallel computations. Now, this unit is integrated with the NEMO software package to perform calculations of the ocean dynamics with DA. The results of the calculations are presented below.

## 3. Experiments on Data Assimilation and Their Analysis

As the observed field, the values of the sea surface height (SSH) were chosen from the Archiving, Validating and Interpolation Satellite Observation (AVISO) archive. They are available on the Internet, here: www.aviso.altimetry.org (accessed on 26 September 2020). The observations were selected along the satellites' tracks whose global scheme is presented in Figure 1 (borrowed from the site www.aviso.altimetry.org).

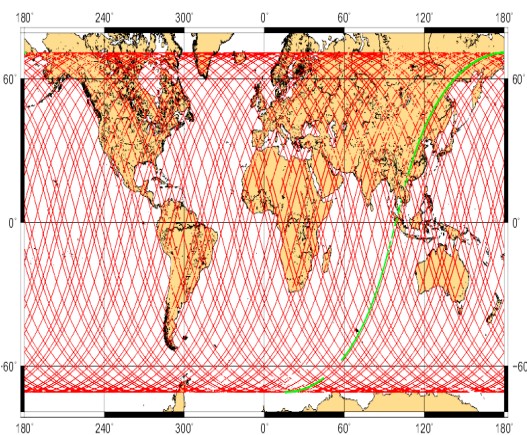

**Figure 1.** Global scheme of satellite tracks.

In our work, we have chosen only the data on the sea surface height in the North Atlantic for July 2013. Assimilation of these data into the NEMO model was carried out by using the GKF method, according to Equations (1)–(4). At that, the localization was introduced as follows: the covariance between two values of the observed SSH was set to be zero, if the distance between the points of

observation exceeded 500 km, i.e., the correlation radius was chosen to be 250 km, which corresponds to 10 grid points. Simulation was performed on a time interval of 14 days. The requirement of theoretical applicability of a DA method is the smallness of the period between two subsequent assimilations as compared with the total time of simulation; in our case, this requirement is satisfied.

Below, we present the results of calculations for the period from July 1 to 15 July 2013, with DA for every day. Preliminarily, after the SPIN UP stage, the model values of SSH for the last 10 years of the preliminary calculation were attributed to these dates. These values were used to calculate the characteristics of the displacement vectors $C_n$ and the covariance matrix $Q_{n+1}$, according to Equations (5) and (6), where $n = 0, \ldots, 14$.

Figure 2 shows the values of SSH calculated by using the NEMO model without DA (control) (Figure 2a), with DA by the GKF method (Figure 2b), and their difference (DA minus control) (Figure 2c) on July 14, 2013, for the particular region of the northern seas (Figure 1).

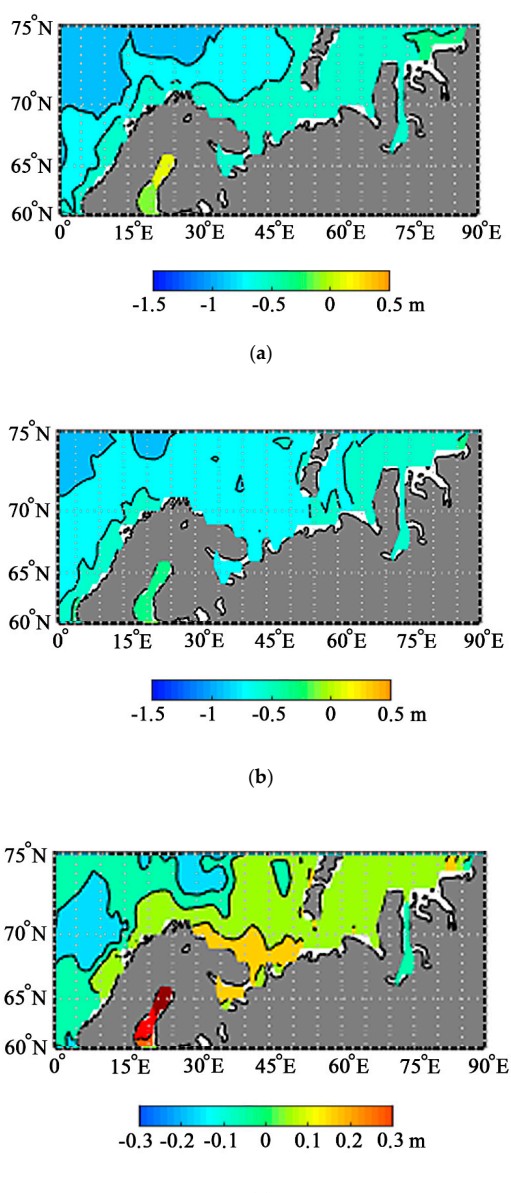

**Figure 2.** Sea surface height field on July 15, 2013: (**a**) calculation without data assimilation (DA); (**b**) calculation with DA; (**c**) difference between sea surface heights (SSHs) in calculations with and without DA.

It is seen in Figure 2a that the ocean surface height rises in the north–northeast direction from −1 m to the north of 70° north latitude to 0.2 m to the east of the Gulf of Ob in the Kara Sea. The model calculation shows quite a continuous and smooth change in the SSH with no distinct kinks or vortices. With DA, the SST field becomes much more chaotic; the zone of minimum values on the northwest of the calculation domain significantly decreases and splits into several zones of local minima; vortex values emerge to the west of the Novaya Zemlya. The zone of SSH maximum of about 0.2 m becomes noticeably smaller and covers the region only to the east of the Gulf of Ob, while to the north of this region, the values of SSH decrease. On the whole, the fields of the model calculation reflect quite well the growth trend along the main currents of this region (the northern branch of the North Atlantic Current [31]). The difference between the assimilated and control values presented in Figure 2c shows the absolute value of corrections and their localization. It is seen that the corrections themselves are not large; they do not exceed 0.3 m in the maximum and have quite a complicated structure. The model noticeably underestimates the SSH to the north of the Kola Peninsula, up to 0.2 m, and to the north of the Novaya Zemlya. On the contrary, the model calculation (control) overestimates the values of the SSH in the Gulf of Ob region and in the region located on the northwest of the calculation domain. It should be noted that the structural peculiarities of the SSH field practically do not change, neither in their localization, nor in the amplitude in Figure 2a,b, which is evidence of the fact that the GKF method does not change the structure of the model fields, and that the NEMO model describes quite well the SSH fields in a large scale. However, for the difference of these fields presented in Figure 2c, the structures of a smaller scale with the amplitude of approximately one half the initial SSH value (the maximum is 0.2 m) are well distinct. This can be explained by some noncoincidence of the centers of the model structures before and after DA, and a small correction of their sizes.

It is interesting to compare the model fields obtained with those plotted by using the real observation data taken from AVISO on 15 July 2013 (Figure 3). We can see in Figure 3 vortex structures in the SSH field on the northwest of the calculation domain, while they are completely absent in Figure 2a; the values themselves of the SSH field in Figure 2b are closer to those of the SSH field presented in Figure 3. The difference between the fields in Figure 2c is in a good agreement with the SSH field in Figure 3.

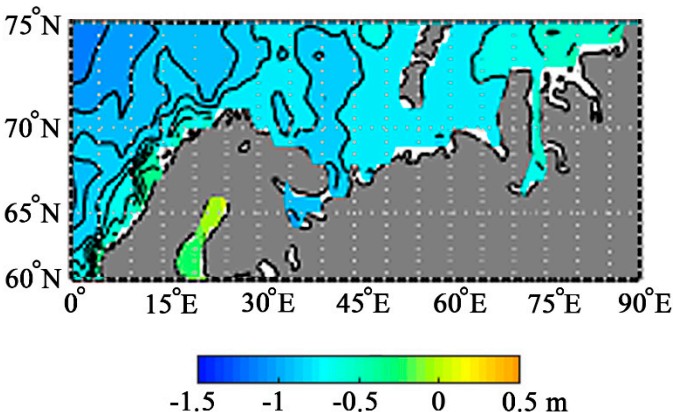

**Figure 3.** Sea surface height field on 15 July 2013, plotted according to the observation data taken from the Archiving, Validating and Interpolation Satellite Observation (AVISO) archive.

It should be noted that in the case of data assimilation, according to Equations (1)–(4), the contribution to the resulting fields with the use of the transfer matrix $K_{n+1}$ according to Equations (2) and (3), consists of the product of the two following components: $\Lambda_{n+1} - C_{n+1}$ calculated on the model grid and the observed component $(H\Lambda_{n+1})^{\mathrm{T}} Q_{n+1}^{-1}$ calculated on the grid of observations; i.e., factorization of variables occurs. This helps us to better understand the DA process itself, separate the regions where the influence of data is the most significant, and quantitatively estimate the influence of particular regions and zones by analogy with the harmonic analysis of the contributions of particular

phases and amplitudes. For this purpose, we have used the decomposition of the matrix $Q_{n+1}^{-1}$ into the empirical orthogonal functions (EOF) [26], i.e., we have represented the vector $(H\Lambda_{n+1})^{\mathrm{T}}Q_{n+1}^{-1}$ in the form

$$(H\Lambda_{n+1})^{\mathrm{T}}Q_{n+1}^{-1} = \sum_{i=1}^{N} \alpha_i \lambda_{n+1}^i e_{n+1}^i, \tag{7}$$

where $\lambda_{n+1}^i e_{n+1}^i$ are the eigenvalues and eigenvectors of the matrix $Q_{n+1}^{-1}$, respectively, $\alpha_i$ are some coefficients. By analogy with harmonic analysis, the vectors $e_{n+1}^i$ can be considered as the phase of the DA process; $\lambda_{n+1}^i$, as its amplitude. The eigenvalues represent the magnitude (the contribution to the energy) of each eigenvectors. Below it is especially shown that the contribution of the first two eigenvectors to the total energy is 99%.

Figures 4 and 5 show the spatial distribution of the two first eigenvectors of the matrix $Q_{n+1}$ for $n$ = 4, 9, 14, since the eigenvectors of the matrices $Q_{n+1}$ and $Q_{n+1}^{-1}$ coincide with each other; their eigenvalues are inverse, i.e., raised to the power –1 with respect to each other.

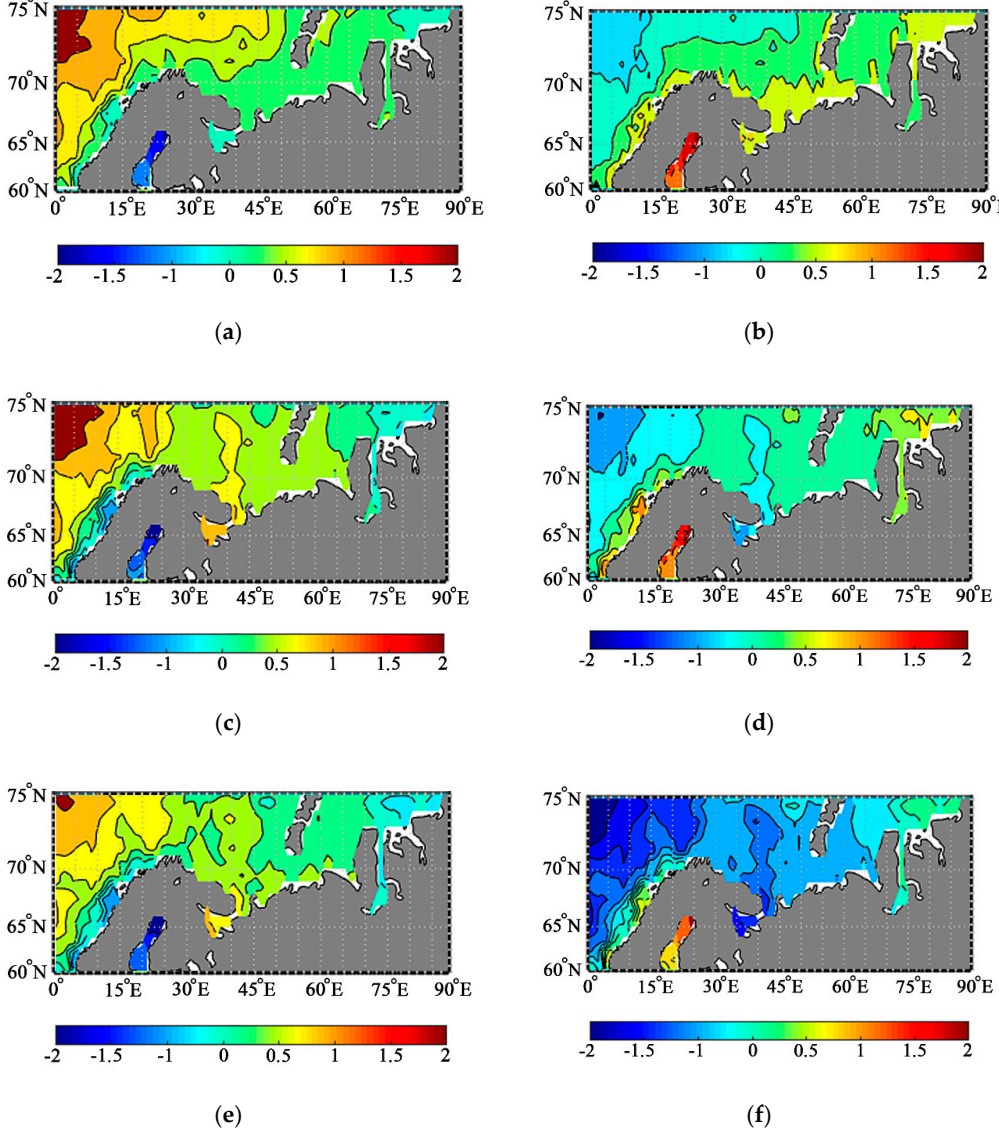

**Figure 4.** Values of the two first eigenvectors $e_{n+1}^1$ (on the left) and $e_{n+1}^2$ (on the right) of the matrix $Q_{n+1}$ at $n$ = 4, 9, 14 (which corresponds to the 5th, 10th, and 15th days of calculation): (**a**) $e_5^1$, (**b**) $e_5^2$, (**c**) $e_{10}^1$, (**d**) $e_{10}^2$, (**e**) $e_{15}^1$, (**f**) $e_{15}^2$.

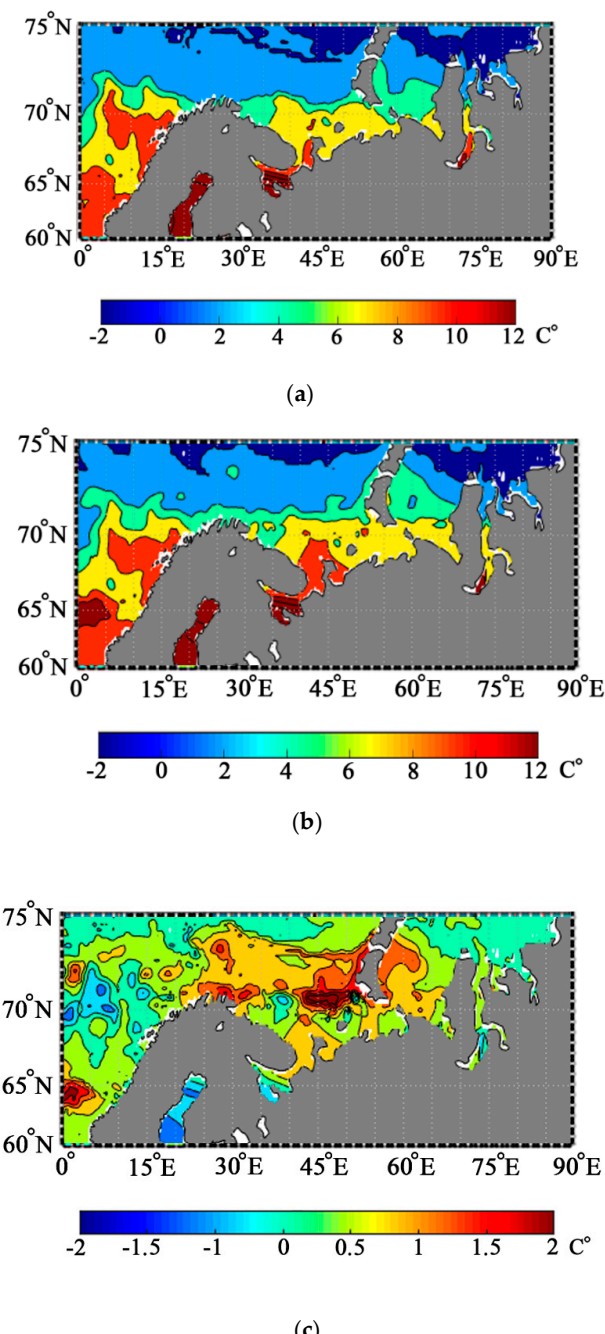

**Figure 5.** Sea surface temperature on 15 July 2013: (**a**) calculation with no DA; (**b**) calculation with DA; (**c**) difference between sea surface temperatures (SSTs) in the calculations with and without DA.

Analyzing these figures, we can come to the following conclusions. At first, it is obvious that the first and second eigenvectors are really orthogonal; i.e., their values have opposite signs and their absolute values are nearly identical. This is seen if we consider the scalar product as a sum of the terms in the decomposition into the basis vectors consisting of zeros and units at the grid points. At second, since we present the values of eigenvectors of the matrix $Q_{n+1}$, while the calculation of corrections involves the matrix $Q_{n+1}^{-1}$, then the maximum eigenvalues of the matrix $Q_{n+1}^{-1}$ will correspond to the minimum eigenvalues of the matrix $Q_{n+1}$ and vice versa. At third, the structure of these eigenvectors is in a very good agreement with the calculated fields after DA and in quite a good agreement with the observations. This is especially noticeable when comparing the results presented in Figures 2b and 4. For example, in Figure 4, we can clearly see a "tongue" of the SSH

values of about −1 m to the north of the Kola Peninsula, which is present in the observation data
and is practically not reproduced in the calculations using the model with no DA (Figure 2a); at that,
in the SSH field with DA, it is noticeable only in the form of small vortices. In additions, we can
note a sea level rise to the northward of 70° north latitudes, which is seen in the observed SSH and is
practically not reproduced in the calculations using the model with no DA; in the calculations with
DA, it is reproduced in a strongly smoothed form; in the eigenvector field, it is reproduced much
more distinctly.

The values of the eigenvalues are presented in Table 1.

**Table 1.** Eigenvalues.

| Date | $\lambda_n^1$ | $\lambda_n^2$ |
|---|---|---|
| 5 July 2013 | 0.97 | 0.002 |
| 10 July 2013 | 0.91 | 0.005 |
| 15 July 2013 | 0.9 | 0.006 |

It is seen from Table 1 that, in essence, the first two eigenvectors make the main contribution to
the resulting fields, while the amplitudes of the 3rd and subsequent eigenvectors are less than 1%.

Next, let us analyze the sea surface temperature (SST) fields. It is clear that a change in the SSH
has a direct effect on the temperature due to the processes of thermal expansion of water. In the NEMO
model, the connection between them is more complicated; however, the principle is the same. Therefore,
the assimilation of the SSH data through the correlation and dynamical link, taken into account in
the GKF data assimilation scheme, will lead directly, i.e., at the moment of assimilation, to a change
in the model SST field as well. Figure 5 shows the SST: before DA (control) (Figure 5a), after DA by
the GKF method (Figure 5b), and their difference (DA minus control) (Figure 5c) for the particular
region of the northern seas in the end of the calculation period, on 15 July 2013.

It is seen in Figure 5 that the corrected temperature fields have some peculiarities in comparison
with the control fields. In both the corrected and control calculations, the SST fields are well structured,
the zonal partition at a resolution of 2 °C is well distinct. It is seen that the "warm water" zone in
the Arctic near the Kola Peninsula in the corrected field (Figure 5b) is somewhat smaller and more
chaotic than that in the calculation with no DA in Figure 5a, where this zone is continuous with a
temperature of about 12 °C. On the whole, in the calculation with no DA, the SST is approximately by
0.5 °C higher than that after DA, i.e., in the calculation with no DA, the values of SST corresponding
to the observed values of SSH are somewhat overestimated as compared to the calculation with DA.
The difference between the SSTs calculated with and without DA presented in Figure 5c is quite a
chaotic field with an amplitude of about −0.5 °C, with negative values to the north of 70° north latitudes
and to the north of Novaya Zemlya, however, with a positive difference of the same order of magnitude
along the coastline of the Kara Sea and in the Baltic Sea.

To estimate if DA is carried out adequately and if the model reproduces correctly the SST,
the results obtained in this study can be compared with the observed SST fields, which are accessible
on the Internet here: www.ostia.noaa.edu.

It is clearly seen in Figure 6 that on the whole, in the calculations using the NEMO model with
and without DA, the main structural peculiarities of the SST in the northern seas are well reproduced.
The values of SST in the calculation without DA are approximately by 1 °C higher than the values
of observation data in the "warm zone", while in the calculation with DA, the values of SST are
approximately by 0.5 °C higher than the observed values. In the calculations with DA, the warmest
zone with the values of temperature higher than 12 °C is somewhat narrower than the observed zone
and is located closer to the coasts. On the whole, we can conclude that assimilation of the SSH data
corrects the SST fields adequately, however, with an insufficient accuracy.

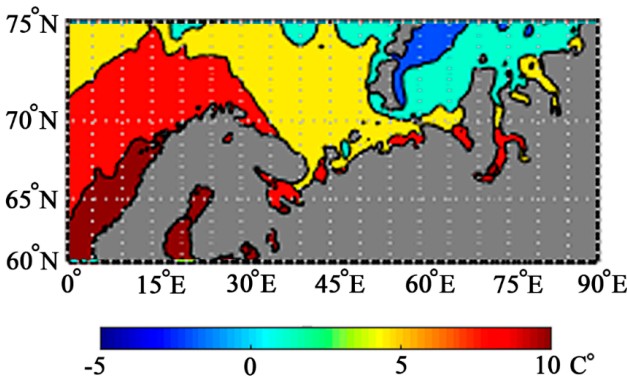

**Figure 6.** Observed SST field on 15 July 2013.

The analysis of currents in the indicated calculation zone has also been performed. Figure 7 shows the results of calculations of the velocity of zonal currents before correction (Figure 7a), after correction (Figure 7b), and the difference between the velocities of currents after and before correction (Figure 7c). We are mostly interested in the zonal component of velocity, since it is known that the main currents pass along the coastal lines of Scandinavia and the Kola Peninsula and are the continuation of the North Atlantic Current.

It is seen in Figure 7a that in the calculation without DA, the velocity of the main current in the west–east direction is about 0.5 m/s; it is localized from the coasts of Norway up to approximately 72°–73° north latitude in the northern direction, and further, up to the coasts of Novaya Zemlya in the eastern direction. The scheme of currents is well known qualitatively [28]; from the quantitative point of view, it is interesting to know the values of the calculated velocities of currents. It is seen from the results of calculation with assimilation of the SSH data shown in Figure 7b that the main current shifts westward up to the coasts of Scandinavia and becomes more chaotic, extending not only across the above-indicated regions, but further northward. The difference of the values of the velocities of currents themselves varies from 0.2 m/s in the regions near the coasts of Scandinavia to −0.2 m/s in the Barents Sea and in the White Sea. Thus, in the calculations with no DA, the values of the currents' velocity are somewhat underestimated on the west and on the north of the calculation region and, on the contrary, overestimated in the Barents Sea and in the White Sea in comparison with the calculations with assimilation of the SSH data. We should especially emphasize a distinct intensification of currents along the coastline of the White Sea and the Kara Sea, namely, more than by 0.2 m/s (Figure 7b), which may be of great importance for calculations of the optimal ways of ships along the Northwest Passage.

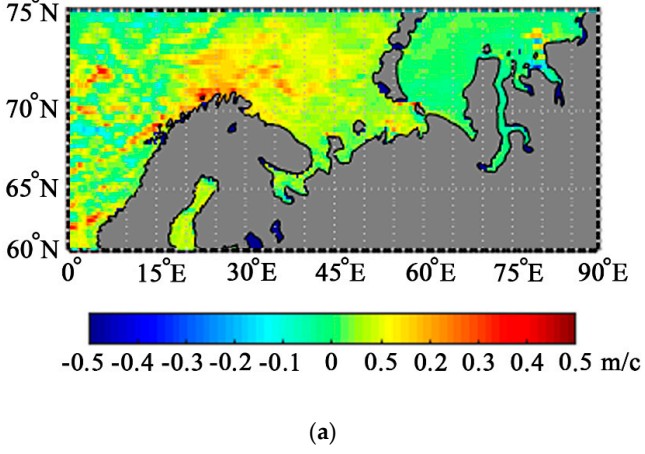

(a)

**Figure 7.** *Cont.*

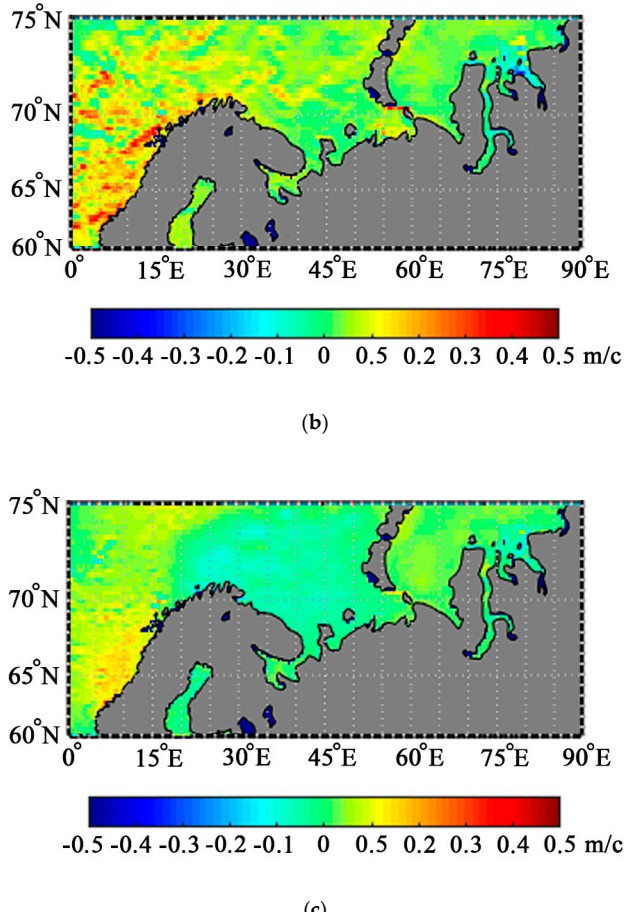

**Figure 7.** Map of the zonal component of the velocity of currents: (**a**) calculation without DA; (**b**) calculation with DA; (**c**) difference between the calculations with and without DA.

## 4. Conclusions

In this paper, we have presented the results of experiments with the NEMO model of ocean circulation and the authors' method of data assimilation GKF. The choice of the NEMO model is conditioned by the fact that this model is widely used in modern physical oceanography for the analysis of the ocean dynamics in different investigations, including simulation of the dynamics of the Arctic region (see, for example, [20]). The choice of the authors' GKF method is justified by some of its advantages over the known EnOI method, which is shown in [1–3].

The results of calculations have shown that the main spatial variability of the calculation characteristics after assimilation of the SSH data are in a good agreement with the localization of currents in the North Atlantic Ocean and in the Arctic zone of Russia. At that, the results of calculations with DA are in a better agreement with the observation data, which is important for practical calculations of currents.

In our further investigations, we plan to extend the field of application of the NEMO model and combine it with local models for the northern seas of Russia with consideration of tides and ice dynamics. In addition, it is planned to perform not only assimilation of the SSH data, but also the data on temperature and salinity, including those from hydrological stations in the Atlantics and northern seas of Russia. It would contribute to the improvement of the quality of analysis and forecast obtained by means of numerical simulations. It may significantly improve the analysis and forecast of the current structure, their dynamics, and their impact on the infrastructure in the Arctic, including oil and gas production platforms, pipelines, coastal protection constructions, etc.

**Author Contributions:** K.B. and A.K. presented the theory; I.S. performed the numerical experiments. All authors have read and agreed to the published version of the manuscript.

**Funding:** This research received no external funding.

**Acknowledgments:** This work was supported by the Russian Science Foundation, project no. 19-11-00076.

**Conflicts of Interest:** The authors declare no conflict of interest. The founding sponsors did not participate in the design of the study; in the collection, analyses or interpretation of data; in the writing of the manuscript, or in the decision to publish the results.

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
