# Peer review of "Spatial–Temporal Variability of the Calculated Characteristics of the Ocean in the Arctic Zone of Russia by Using the NEMO Model with Altimetry Data Assimilation"

_jmse, doi:10.3390/jmse8100753_

Round 1

Reviewer 1 Report

Review of the paper “Spatial-Temporal Variability of the Calculated Characteristics of the Ocean in the Arctic Zone of Russia by using the NEMO Model with Altimetry Data Assimilation”

This paper illustrates a modelling experiment performed in the Arctic zone of Russia by using the NEMO model with assimilation of SSH data based on the GKF (Generalized Kalman filter) method previously developed by the authors.

The results are shown on a short period, 15 days (1 to 15 July 2013, page 6 line 211), while usually these kind of forecasting/hindcasting systems (model + data assimilation scheme) are tested on periods of years, or several seasons. Moreover, some of the presented results seem poor, as the uncertainty around 0.2 m/s in some areas (page 14, lines 347-352) probably results in the uncertainty of 30-40% at least on the estimate of the true current.

The methodology is well illustrated and the followed procedure can be useful for further studies. For this reason, I think that the paper deserves publication.

There is one major criticism:

  • In the abstract (line 26): The calculation results are analyzed and their spatial-temporal variability over a time scale of 14 days is studied” should be substituted by ‘over a period of 14 days.’ In a 14-days time scale analysis results, errors etc…are analyzed on sub-periods of 14 days, while here only one interval of 14 days is considered;

and one minor criticism:

  • The use of the term ‘oceanology’, in several parts of the paper, could be substituted by the term ‘physical oceanography’, as also suggested by the title of the paper.

Author Response

Response to Reviewer 1 Comments

Point 1: The results are shown on a short period, 15 days (1 to 15 July 2013, page 6 line 211), while usually these kind of forecasting/hindcasting systems (model + data assimilation scheme) are tested on periods of years, or several seasons. Moreover, some of the presented results seem poor, as the uncertainty around in some areas (page 14, lines 347-352) probably results in the uncertainty of 30-40% at least on the estimate of the true current.

Response 1: We agree that to test completely the GKF method, a period of 15 days is not sufficient. However, it should be noted that the GKF method has been tested early for a longer period [1]; here, we solve the DA problem for a given region for a period of 15 days. In addition, it was pointed out that all conditions which are necessary for GKF application are satisfied.

On page 14, the velocity variability is indicated rather than its accuracy.

Point 2: In the abstract (line 26): The calculation results are analyzed and their spatial-temporal variability over a time scale of 14 days is studied” should be substituted by ‘over a period of 14 days.’ In a 14-days time scale analysis results, errors etc…are analyzed on sub-periods of 14 days, while here only one interval of 14 days is considered;

Response 2: “over a time scale of 14 days is studied.”   has been changed by “over a time period of 14 days is studied.”

Point 3: The use of the term ‘oceanology’, in several parts of the paper, could be substituted by the term ‘physical oceanography’, as also suggested by the title of the paper.

Response 3: “oceanology” has been replaced by “physical oceanography”.

Authors thank to the Reviewer for useful comments, which helped to improve the paper.

Reviewer 2 Report

This article propose an interesting model for the spatial-temporal variability of the calculated characteristics of the ocean in the Arctic zone of Russia. Several techniques are involved in the model including AVISO, GKF and EOF to accurately predict the spatial-temporal variability of the ocean of 14 days. Results of the sea surface height field and temperature without DA are compared with DA. This paper overall presents a quality work with reasonable amount of results and analysis. It is also well-structured and written in quality English. Hence, it is recommended to be accepted with the following comments: 1. Although this work is a continuing study after previous references [1-3], it is still worthwhile to provide some relavent background at the opening for the introduction. Please revise the first paragraph to briefly describe the significance of this work in a few sentence. 2. In Abstract, line 17, the sentence "The spatial-temporal variability ..." is too long. Try to break down this type of sentence to two or more. Please also make changes with similar occassions within the manuscript. 3. In Table 1, what does the eigenvalues represent and how does it influence to the sea surface temperature? There can be more in-depth description here so that the readers can understand the key variables in this model. 4. The Conclusion can be strengthen to include more key factors, findings and important figures indetified from this proposing model.

Author Response

Response to Reviewer 2 Comments

Point 1: Although this work is a continuing study after previous references [1-3], it is still worthwhile to provide some relavent background at the opening for the introduction. Please revise the first paragraph to briefly describe the significance of this work in a few sentence

Response 1: On the pages 1-2 the extra text was added.

Point 2: In Abstract, line 17, the sentence "The spatial-temporal variability ..." is too long. Try to break down this type of sentence to two or more. Please also make changes with similar occassions within the manuscript.

Response 2: The first sentence in Abstract is divided into two sentences.

Point 3: In Table 1, what does the eigenvalues represent and how does it influence to the sea surface temperature? There can be more in-depth description here so that the readers can understand the key variables in this model.

Response 3: The eigenvalues represent the amplitude (the contribution to the energy) of each eigenvectors. It is indicated specially that the contribution of the first two eigenvectors to the  total energy is 99%. This explanation is added to the text (page 9).

Their influence on the sea surface temperature is not known, since these values influence  directly on the sea surface height. The cross-covariance between the sea surface temperature and the sea surface height requires a special investigation which is out of the scope of the paper.  

Point 4: The Conclusion can be strengthen to include more key factors, findings and important figures idetified from this proposing model.

Response 4: On the page 14, the extra text was added.

Authors thank to the Reviewer for useful comments, which helped to improve the paper.